



# Digitization and calibration of historical solar absorption infrared spectra from the Jungfraujoch site

Jamal Makkor[1], Mathias Palm[1], Matthias Buschmann[1], Emmanuel Mahieu[2], Martyn P. Chipperfield[3], and Justus Notholt[1]

[1]University of Bremen, Institute of Environmental Physics (IUP), Bremen, 28359, Germany
[2]Institute of Astrophysics and Geophysics, UR SPHERES, University of Liège, Liège, 4000, Belgium
[3]School of Earth and Environment, University of Leeds, UK

**Correspondence:** Jamal Makkor (makkor@uni-bremen.de)

**Abstract.** This study describes the digitization and calibration of historically significant solar absorption spectra recorded at the Jungfraujoch International Scientific Station in the 1950s. Using a homemade Pfund-type grating spectrometer, these spectra were recorded on paper rolls to study the solar spectrum which was then used to compile a solar atlas between 2.8 and 23.7 microns (~421 to $3571 cm^{-1}$) that later contributed to the development of the HITRAN database. We now digitized these old

spectra to make them available for atmospheric studies. Our approach involves image processing techniques, including colour masking for digitization and peak detection for accurate wavenumber calibration against a synthetic spectrum.

We also developed a validation method by re-digitizing degraded FTIR spectra to the same resolution as the old spectra to evaluate the digitization accuracy. Furthermore, we studied the influence of line thickness on the digitization error.

The number of spectra transformed into a machine-readable format is 108 (freely available for download), with an average

digitization error of 1.55% and a wavenumber shift standard diviation of $0.075 cm^{-1}$. These digitized and calibrated spectra now offer a valuable resource for atmospheric studies, providing essential historical data for atmospheric research. This work not only helps to preserve scientific heritage but also enhances the utility of historical data in contemporary research.

## 1   Introduction

The Jungfraujoch station is located in the Swiss Alps (46.55N, 7.98E) at an altitude of 3580m, on a saddle between the Mönch

(4107m) and the Jungfrau (4158m) summits. The Sphinx observatory, built in 1931 (ISS, 1931), focuses on atmospheric studies (among many other scientific disciplines) and provides continuous measurements of various species using state-of-the-art technologies (Zander et al., 2008). Its strategic location at high altitude, coupled with minimal interference from pollution and water vapour, renders it an optimal setting for such studies.

The first atmospheric measurements on this site started in the 1940s using grating based spectrometers, including the one-metre

focal length Pfund-type prism-grating instrument described in this paper, which operated between 1950 and 1951. Later on, from the late 1950 to 1980 a seven-metre focal length Ebert/Fastie-type prism grating spectrometer designed at the University of Liege (U.Liege) was installed dedicated to near-UV near-IR studies extending between (300 to 1200nm). In 1984, the observatory became aware of the prevalence and efficiency of Fourier Transform Infrared based spectrometers (FTIR) com-



pared to their grating counterparts, and therefore augmented its capabilities by integrating a homemade FTIR spectrometer.

Then starting from the early 1990s, an FTIR manufactured by the German company Bruker (IFS 120HR) and modified by the U.Liege team was added and operated in parallel with the homemade spectrometer. The homemade FTIR was later retired in 2008 (Zander et al., 2008), while the Bruker instrument is still in use.

The solar atlas that resulted from the spectral rolls and produced by Migeotte et al. (1956), extending from 2.8 to 23.7 microns, has been a historically scientific achievement and a valuable resource for scientists for decades. It includes spectra of

atmospheric gases such as carbon dioxide, ozone, and water vapour etc, covering a broad range of wavenumbers. The process of identifying spectral features in this atlas was carried out by Migeotte from 1950 to 1956. Researchers from various fields, including climate science (Zander et al., 1994), atmospheric physics (Ehhalt et al., 1983), and chemistry (Murcray et al., 1978), have utilized this information to understand the behavior and interactions of these gases in Earth's atmosphere. Migeotte used the instrument to detect CO at the top of the Alps, a location previously assumed to be unpolluted (Migeotte and Neven, 1950).

Zander et al. (1989) then calculated the total column of this gas using these spectra. Additionally, in 2008 Zander et al. (2008) investigated the existence of dichlorodifluoromethane ($CCl_2F_2$ commonly known as CFC-12 or Freon12™) in the old spectra through visual comparison with high-resolution spectra.

Despite their potential value, the spectra have not been fully utilized by the scientific community, primarily because they are not available in a digital format that can be easily shared. In this work we have developed a method to digitize and calibrate

these spectra to the proper wavenumber range. To achieve this, a digitization/calibration software was developed. This software transformed the original paper spectra into a machine-readable and universally accessible ASCII format. The digitization process employed image processing techniques to detect plotted lines and extract the spectrum, while the calibration phase involved curve-fitting the produced spectrum (pixel-mapped) to a synthetic one covering well known atmospheric lines on a known wavenumber scale.

## 2  Methods


The paper spectra were scanned using an Epson A3 (WORKFORCE DS-50000 (EDC)) professional scanner. The resulting high-resolution uncompressed TIFF files were then preprocessed, corrected for any misalignment, and combined into a single file. To digitize and calibrate these spectra, a software was developed using Python and the Tkinter graphical library. The spectra were stored in a machine-readable format not only for easier analysis but also to preserve them for posterity. The

Pfund-type grating that was used to record these spectra (shown in Figure 1a and illustrated in Figure 1b) used a thermocouple as a detector (Table 1). Thermocouples have commonly been used in the past as detectors of infrared radiation due to their cost-effectiveness, simplicity, and wide temperature range. However, a significant drawback of thermocouple-based spectra is the high level of thermal noise. This causes these types of detectors to have a lower signal to noise (S/N) ratio compared to the recent ones. Additionally, an important quantity calculated during the extraction of the spectra is the solar zenith angle

(SZA) which is often used in atmospheric retrievals. The SZA was calculated based on date, time and location. However, the





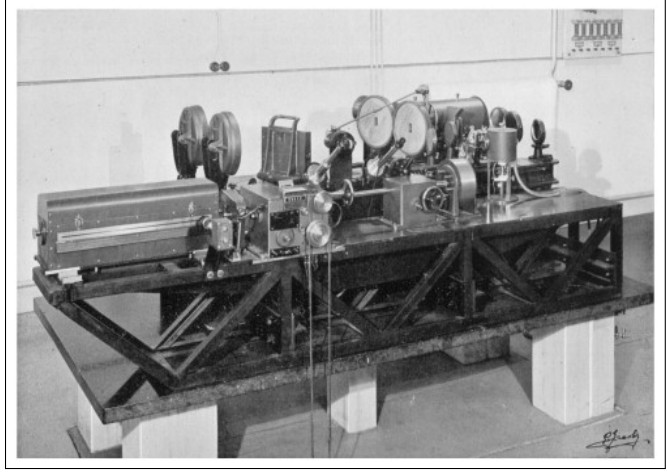
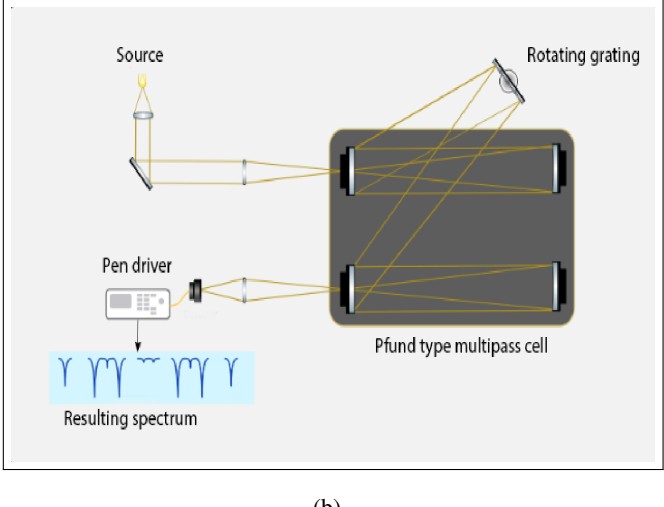

(a)                 (b)

**Figure 1.** The spectrometer used to produce the historical spectra. **(a)** The instrument was a home-made one-meter focal length Pfund-type grating spectrometer. It was described by Migeotte when he reported the solar atlas in 1956 (Photograph from Zander et al. (2008)). **(b)** A simplified schematic of the Pfund-type grating spectrometer (adapted from Miller and Thompson (1949)).

**Table 1.** Technical specifications of the Pfund-type Grating Instrument and the produced spectra.

| Parameter | Value |
|---|---|
| Operation years | 1950 to 1951 |
| Resolution [cm$^{-1}$] | 0.12 to 0.4 |
| Signal to noise ratio | 40 to 80 |
| Spectral range (at various intervals) [$cm^{-1}$] | 520 to 3565 |
| The instrument lineshape (ILS) function | $sinc^2$ |
| Apodisation function | Triangular |
| Detector | Perkin Elmer Thermocouple |

measurements using the grating took a relatively long time, up to 2.5 hours (1.5h average recording time). This influences the value of SZA and subsequently the airmass. This needed to be taken into account as well when producing the spectra.

## 2.1 Digitization

This section describes the method used to extract spectral data from the spectral rolls. They were marked with additional

information, such as the date, start and end recording time, high and low wavenumber, and occasionally surface temperature





and humidity (which were gathered from notes on the papers during the scanning process). The recorded lines were plotted using red ink, and an algorithm was developed to detect these lines using image processing.

### 2.1.1 Scanning and Preprocessing

The spectral rolls presented a range of conditions, some of which contained annotations from various scientists who had previously engaged in the analysis of these spectra. Others were repaired using adhesive tape. Notably, the presence of tape did not impede the digitization process, as its coloration was distinct from that of the spectral lines. A critical preparatory step involved ensuring that the spectral rolls were aligned as horizontally as possible prior to scanning, a measure taken to mitigate potential distortions. However, very small misalignment that cannot be observed by the naked eye, proved to also have an effect on the total digitization error. This was observed during the validation phase (a slope in the difference between calibrated and original spectrum caused by a small rotation in the image). To mitigate this, an automatic alignment and cropping algorithm was also developed to correctly adjust the images.

The employed scanner offers A3 format capabilities and a resolution of 600 dpi for uncompressed images, which were notably large in file size (thus offering higher digitization accuracy, but needing longer processing periods). However, due to the extended length of some rolls, it was necessary to perform multiple scans to adequately capture the entire spectral range of a single roll. Subsequently, these individual scans were automatically merged into a full spectrum using image editing software (GIMP (2019) and Hugin (2020)). Once the spectral images were obtained, they were processed for plot extraction using the designed algorithm.

### 2.1.2 Digitization algorithm

Figure 2a shows the GUI (Graphical User Interface) used to digitize the paper spectra. The digitization algorithm used to turn the plots in the image files into a readable text format runs as follows:

– Image reading and colour space conversion:
   Python OpenCV's library (Zelinsky, 2009) is used to read and process the images. The colour space of the image is converted from BGR (Blue, Green, Red) to HSV (Hue, Saturation, Value) to facilitate colour-based filtering.

– Straightening and cropping:
   Before the digitization an automated straightening step was applied to the image. This is achieved by fitting a rectangle to the image and then applying a rotation matrix to it. The angle used for this operation is calculated between the fitted rectangle and the horizontal line in the image canvas. Following this, the resulting image is then cropped.

– Colour Filtering:
   A mask is created to filter out specific colour ranges, in this case, using a minimum and maximum threshold. This step isolates parts of the image relevant to the spectral data, this threshold can be adapted to detect any given colour. Figure 2b shows the spectrum taken from the detected line. The visible discontinuities in the spectrum come from the ruled lines which interfere with the colour detection.





- Binary Thresholding:

  The masked image is converted to greyscale. Following this, a binary threshold is applied to create a black and white
  image, further isolating the spectral lines (Figure 2c).

- Spectral Data Extraction: The spectral data is extracted by calculating the mean of the pixel values along the vertical axis
  of the columns of the binary image then linearly interpolated to account for any missing pixels. This step translates the
  visual spectrum into numerical values.

Since the line has an inherent thickness coming from the pen, the mean of the thickness of the detected line was taken as the
resulting spectrum. However, the true spectrum lies in between the border of this line and the thickness needs to be accounted
in the calculation of error. The standard deviations of the detected plot points are shown in Figure 2d. The calculated mean
deviation is about 9.3 pixels, taking into account an image resolution of 600 dpi.

This gives us a relative mean digitization error of about: $\epsilon = \frac{9.3*100}{600} = 1.55\%$

### 2.1.3 Accompanying metadata

In addition to the spectral data, housekeeping information was also recorded and stored alongside the generated spectrum. This
data encompassed the lowest and highest wavenumber, solar zenith angle range, and airmass. When documented on the rolls,
temperature and humidity values were also included.

## 2.2 Calibration

The digitized spectrum, saved only as pixel positions, lacks a definition in the correct wavenumber range. After digitization,
calibration to this range is necessary. This was achieved by first producing a synthetic spectrum with a known wavenumber
range. Then after choosing appropriate calibration points, fitting the digitized data-points to the synthetic spectrum thus defining
the appropriate wavenumber scale.

### 2.2.1 Synthetic spectrum preparation

The synthetic spectrum was generated using the SFIT4 forward model, leveraging known spectroscopy from the High-Resolution
Transmission Molecular Absorption Database (HITRAN) (Gordon et al., 2017) and specific instrumental parameters, such as
the apodization function and resolution (https://wiki.ucar.edu/display/sfit4/). The synthetic spectrum was calculated using an
averaged resolution of $0.25\,\mathrm{cm}^{-1}$ and a triangular apodization according to the properties of the Pfund spectrometer (see Table
1).

The choice of the apodization function for the interferograms is connected to the instrumental properties of a grating spectrom-
eter, since for a diffraction limited resolution, where the image of the spectral line is no longer the same shape as the entrance
slit but controlled by the diffraction pattern, the instrumental line function in the spectral domain corresponds to a $sinc^2$ func-





(a)

(b)

(c)

(d)

**Figure 2.** Digitization of a scanned old spectrum. **(a)** The digitized spectrum (in blue) extracted from the pen plotted line (in red). The circled zoomed in view shows sudden peaks that appear throughout the spectra. **(b)** Resulting image after applying the colour detection mask. this method allows the detection of a line despite the handwritten annotation in the image. **(c)** The resulting image after binary threshholding of the grayscale image. **(d)** Histogram of the standard diviations of each digitized point in the spectrum. The mean of the calculated deviation is about 9.3 pixels.

tion (Griffiths, 2002) (therefore justifying the use of a triangular apodization in the interferogram domain when producing the

125    synthetic spectrum).

The spectral lines of the synthetic spectrum do not perfectly match the calibrated spectrum due to line broadening and line strength differences, however, their center positions remain unchanged allowing for the calibration.

The synthetic spectrum was produced to cover the whole range of the spectral rolls (from $500cm^{-1}$ to $5000cm^{-1}$) using a fixed $0.25cm^{-1}$ resolution. However, it is worth mentioning that for a grating spectrometer the resolution is waevnumber dependent

130    and the averaged resolution was thus used for practical reasons. This spectrum is then read and plotted by the software to be used in the subsequent calibration step.





### 2.2.2  Calibration methodology

The calibration algorithm used to map the digitized spectrum to the correct wavenumber range is performed as follows:

- Metadata such as date, time, minimum and maximum wavenumber and location coordinates are read from a configuration file saved during the spectrum extraction phase (see Section 2.1.2). They are then used to calculate the solar zenith angle, azimuth and various other parameters which are saved in the final spectrum or provided in the accompanying housekeeping file.

- Both the synthetic spectrum and the digitized pixel values are loaded side by side (see Figure 3a). The synthetic spectrum is shown at the appropriate wavenumber range read from the configuration file, and the peak detection boundary is defined (horizontal blue line).

- The relative maxima search method is the peak detection algorithm used here to identify calibration points. This method identifies a data point as a peak if it is greater than its neighbors on both sides, effectively pinpointing the maximum values of each spectral line. While this approach is typically effective, there may be instances where the peak detection identifies more data points in either the digitized spectrum or the synthetic one. In such cases, manual removal of misidentified peaks is also possible.

- In case the peak detection algorithm fails (if the paper based spectrum is too noisy for example), it is possible to manually select calibration points.

- The calibration points are then saved and used to curve fit the digitized data points to the synthetic spectrum using least square fitting method and a $2^{nd}$ degree polynomial model (which was sufficient to give a good fit without risking overfitting at higher degree ones). The spectrum is then saved as a text format with the appropriate wavenumber range, ready for use.

### 2.2.3  Choice of curve fitting function for wavenumber calibration

In our analysis of curve fitting models, we assessed both $\chi^2$ values (which represents an insightful overview of the goodness of the fit) and the standard deviation in wavenumber shift. The $2^{nd}$ degree polynomial emerged as the optimal model, balancing fit quality and wavenumber shift precision. While the $3^{rd}$ degree polynomial showed a lower $\chi^2$ value (more than half of that for the $1^{st}$ degree polynomial), it suffered from a higher standard deviation in wavenumber shift (more than twice that of the $2^{nd}$ degree polynomial). The $1^{st}$ degree polynomial, though simpler, had the highest $\chi^2$ value and a relatively high standard error, indicating a poorer fit and less reliability. Given these observations, the $2^{nd}$ degree polynomial was selected for its better balance between a lower $\chi^2$ value (only slightly higher than the $3^{rd}$ degree polynomial) and a significantly more stable standard deviation of the wavenumber shift.

Figure 4 shows a plot of the wavenumber versus the pixel calibration points detected by the peak detection algorithm and fitted to a $2^{nd}$ degree polynomial and the mean values of different statistics for three polynomial models are detailed in the Table 2.

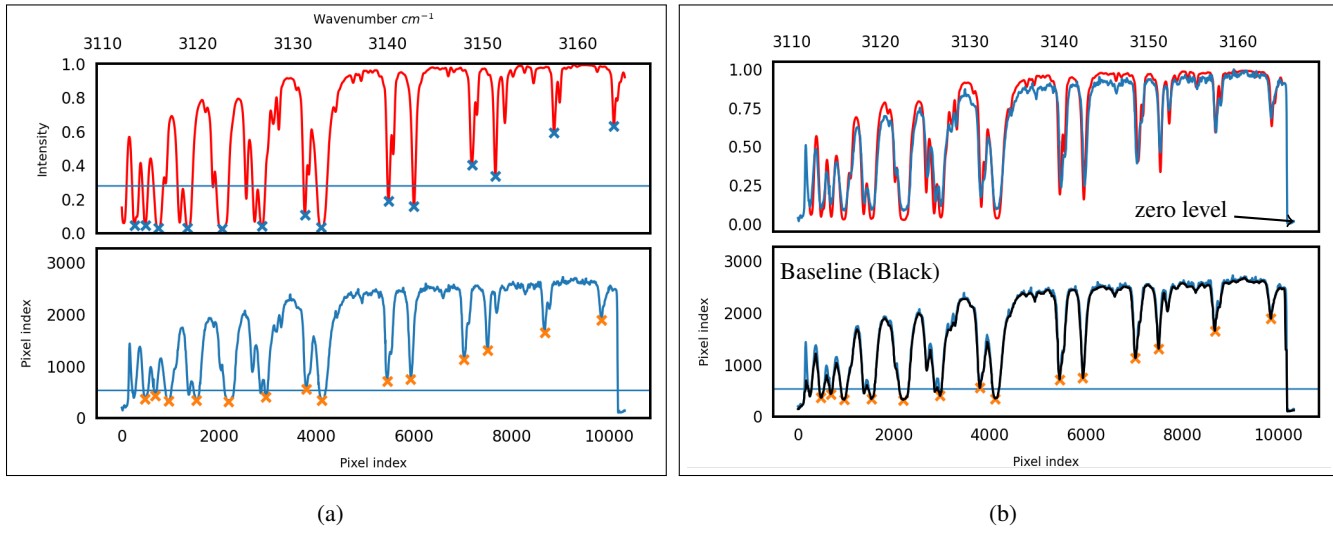

(a)

(b)

**Figure 3.** Calibration of an old spectrum. **(a)** The lower plot shows the digitized spectrum and the upper plot shows the normalized synthetic spectrum (generated using SFIT4). Both spectra show the detected calibration points by using a chosen detection limit (blue horizontal line). **(b)** The lower plot shows the digitized (blue) spectrum with its baseline (black). The upper plot shows both the calibrated (blue) and simulated (red) spectra.

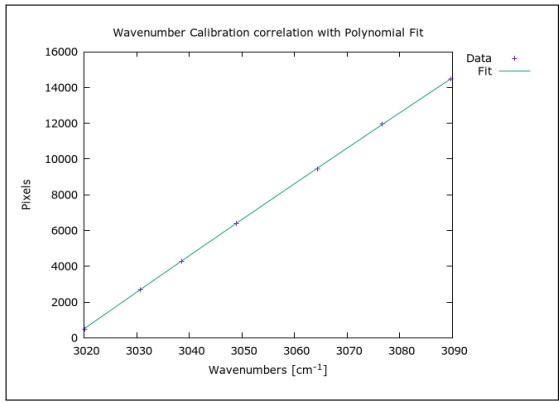

**Figure 4.** Pixel calibration to the correct wavenumber fitted to a $2^{nd}$ degree polynomial.

The spectra displayed abrupt vertical spikes, or upticks, throughout the recording, the origin of which remains unclear (see the zoomed in circle in Figure 2a). These upticks posed challenges in saving a normalized spectrum at the accurate zero level.

Fortunately, the zero level was consistently indicated in the recorded spectra. To mitigate this, each spectrum was fitted to a baseline, effectively ignoring sudden peaks and determining the maxima and minima for normalization. Each normalized spectrum was saved as an ASCII text file.

The error at the zero level can be explained by the fact that even in the absence of a signal (i.e. no solar radiation) at the detector there are still some voltage fluctuations (noise) recorded by the pen on the paper. This adds some additional uncertainty to





**Table 2.** Comparison of polynomial models used in curve fitting the digitized spectrum.

| Statistic | N>3 Polynomial | N3 Polynomial | N2 Polynomial | N1 Polynomial |
|---|---|---|---|---|
| $\chi^2$ | No Fit | 0.03 | 0.05 | 0.1 |
| Wavenumber Shift Std $(cm^{-1})$ | - | $\pm 0.165$ | $\pm 0.065$ | $\pm 0.127$ |

retrievals performed by the spectra. This offset error was estimated by Zander et al. (1994) to be around 3% (referred to as zero transmission offset).

During the extended recording durations, the SZA shows significant variations, potentially leading to inaccuracies in retrieval values. The `pvlib` package was utilized to calculate the SZA array at different recording times, the results were compared against NOAA's solar zenith angle calculator (https://gml.noaa.gov/grad/ antuv/SolarCalc.jsp).

**3   Method validation**

The original spectra were produced in the absence of comparable datasets or spectra for validation, requiring unique validation approach for our digitization and calibration method. To verify the efficacy of the digitization method we utilized high-resolution FTIR spectra from the Jungfraujoch site which were first artificially lowered to a resolution of $0.25 cm^{-1}$ by truncating the Fourier transform of the high-resolution spectrum to match the required resolution. Subsequently, the spectra
were subjected to triangular apodization (the reason for this is explained in Section 2.2.1), a process facilitated by the Bruker Opus Software (OPUS, 2017).

Following this, we plotted the low resolution FTIR spectrum (using a similar pen colour) on a background with a texture comparable to the original paper used to produce the old spectra. We also added similar annotation to this background.

This spectrum was then printed on physical paper to try and replicate the effects of printing, like the physical aging of the paper,
and non-uniform colour of the plotted line. It was then rescanned back to a digital format at 600 dpi resolution. Finally, the resulting image was digitized and compared to the original spectrum. Figure 5b illustrates a comparison between an example of the artificially degraded high resolution (HR) FTIR spectrum and the resultant digitized spectrum. The residuum in this case is about 1.5%. This could be attributed to multiple reasons. One of which could be caused by printer accuracy, where printers might not reproduce the colours and details of the original spectrum. Another reason could also be attributed to scanning error
where the colour might be modified by the scanner.

The observed increase in residuum especially at deeper lines can be explained by an amplified error at higher slopes, where small discrepancies between the digitized spectrum and the original one can produce larger differences thus influencing the error. This can also be observed in the digitization standard diviation before calibration.





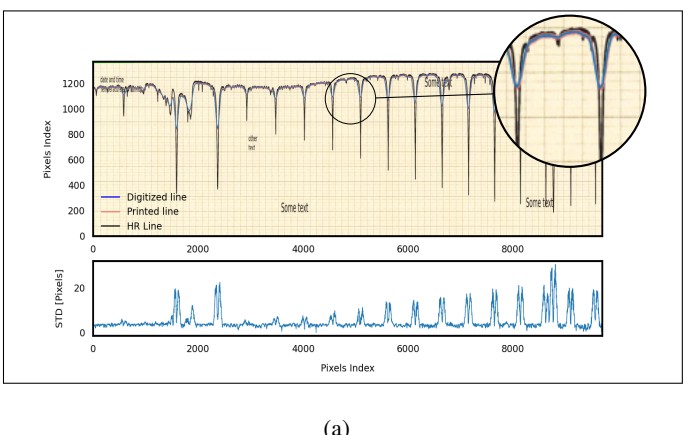

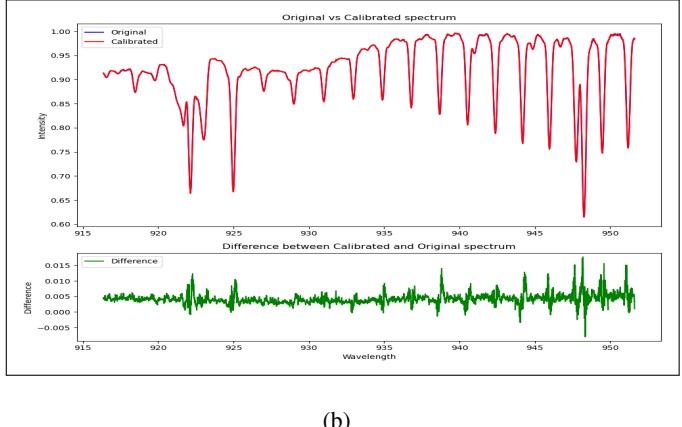

(a)                                              (b)

**Figure 5.** Lower resolution FTIR spectrum printed on paper and scanned. **(a)** The high-resolution (HR) FTIR spectrum (in black) was reduced to a lower resolution and appodized using a triangular function (plotted in red) before being digitized (the digitized line in blue is plotted over the low resolution line). **(b)** Comparison between the original spectrum and the digitized one is showing a good alignment. A digitization residuum of 1.5% is observed.

## 3.1 Influence of line thickness

The original ISSJ spectra had an inherent line thickness coming from the pen. The line thickness of the printed line has an influence on the digitization error. Therefore, quantifying the efficiency of the digitization algorithm at different line thicknesses is crucial.

Figure 6 displays the same spectrum (zoomed in view) digitized at three different line thicknesses: the same as the old paper spectrum, half, and double the thickness), calibrated and then compared to the original spectrum. The spectrum was produced

at a fixed spectral resolution of $0.25cm^{-1}$.

In our analysis, we observed a notable impact of line thickness on the accuracy of spectra digitization, as demonstrated in Figure 6. Specifically, we found that the discrepancy between the original spectrum and its digitized counterpart increases with the thickness of the plot lines.

This result can be attributed to an inherent limitation in the digitization process when handling thicker lines. In the case of

spectral lines with significant thickness, the digitization algorithm is designed to calculate the mean position of the detected line. However, the actual location of the true spectral line within this thicker plot remains ambiguous. When a plot line is sharply defined and narrow, the original spectrum position is usually closer to the mean digitized line. Conversely, as the line thickness broadens, the line's precise location becomes increasingly obscured, merging into the overall thickness of the line. This leads to greater uncertainty in determining the exact position of the spectral data during digitization.

Therefore, thicker lines introduce an additional layer of uncertainty in this digitization process. The algorithm's reliance on averaging across the line's width means that any deviation from the true line position within the thickness of the line directly





contributes to increased digitization error. Consequently, the accuracy of digitization is inversely related to the thickness of the spectral lines, with thicker lines resulting in a higher likelihood of deviation from the true spectral data.

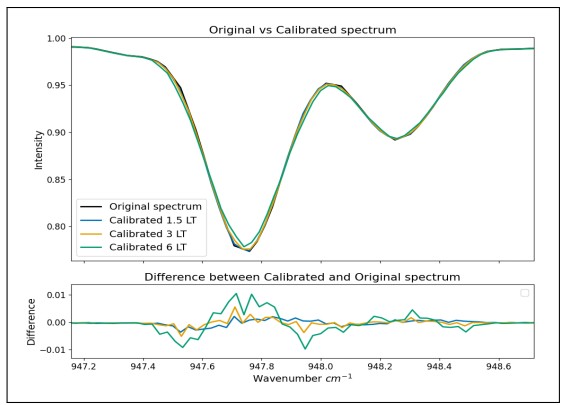

**Figure 6.** A zoomed in view of an original degraded FTIR spectrum and the redigitized version of it at different line thicknesses (LT). A 3LT corresponds to 25 pixels thickness.

## 4 Error Study

The spectral lines produced by the grating spectrometer have an inherent thickness caused by the use of the recording pen. During the digitization process, the mean of the detected line for a given pixel array was taken. The digitization process of the spectral lines results in an estimated 1.55% error due to the digitization alone.

The $2^{nd}$ degree polynomial proved a better choice for curve fitting the spectra, although having an overall higher $\chi^2$ value than the $3^{rd}$ degree one, it has a lower wavenumber shift STD. Additionally, when calibrating the digitized spectrum to the synthetic

wavenumber-mapped spectrum, we need to account for wavenumber fitting shift error of about $\pm 0.065\,\mathrm{cm}^{-1}$. The difference between the detected peak (or minima) of the line and the true peak needs to be quantified as well. The true peak is the center of the line calculated from spectroscopic measurements by HITRAN and the error coming from the calibration using peak detection can be calculated as $\epsilon_{peak} = \nu_{true} - \nu_{detected}$. This is done by comparing the line center from HITRAN lookup table to the peak detected wavenumber. After fitting the spectra and comparing the wavenumber difference an additional estimated

error of $0.01 cm^{-1}$ was calculated.

The error from wavenumber calibration can be expressed as follows $\epsilon_{cm^{-1}} = \sqrt{\epsilon_{peak}^2 + \epsilon_{fit}^2}$. This error needs to be added to the spectroscopic error from the chosen linelist when using these spectra. It was shown that the line thickness plays a role in influencing the error. Where the digitization error doubles when doubling the line thickness.

Additionally, it is worth mentioning that this digitization method only works if the colour of the plot on the paper is different

from the rest of the paper (including annotations). Otherwise, in case there are artifacts of the same colour as the spectrum, the digitization will fail. It is then imperative, when possible, to remove any artifacts that might cause the algorithm to misidentify lines.



## 5   Summary and Conclusion

This paper describes the process of digitizing historical atmospheric spectra from the 1950s, originally recorded at the Jungfrau-
joch International Scientific Station using a Pfund-Type grating spectrometer. The digitization utilized high-resolution scanning
and a colour masking technique in image processing to accurately capture the spectral data. Calibration was achieved using a
synthetic spectrum based on HITRAN data, aligning the digitized spectra with the correct wavenumber range at an estimated
digitization error of about 1.55% and a wavenumber calibration STD of about $0.075cm^{-1}$. The study revealed a relationship
between the plotting line thickness and the corresponding digitization error, which unsurprisingly increased with increasing
line thickness.

In conclusion, the successful digitization and calibration of these historical spectra have preserved valuable scientific data,
facilitating future atmospheric research and comparisons with modern datasets. This work will hopefully contribute to the field
of atmospheric science and potentially other relevant fields.

### Data records

The data records are saved as individual spectra (ranging from $520cm^{-1}$ to $3565cm^{-1}$), each containing relevant information
for data analyses. Each spectrum has a header that contains the minimum and maximum wavenumber, the number of points,
and the resolution, among other details (SZA, apodization type etc...). Accompanying the spectra is a housekeeping file that
contains additional data. To facilitate the visualization of the digitized/calibrated spectra, a web portal was created using
the Python Flask framework (Grinberg, 2018). The produced spectra can be visualized on the web portal https://iup.uni-
bremen.issj.spectra.makkor.de . The files are saved in plain text format, making them easily accessible to users. Additionally,
the solar zenith angle (SZA) and airmass ranges are saved as a text array.

*Code availability.*   A snapshot of the digitization and calibration code can be downloaded from the following source https://zenodo.org/records/11204115.
The web portal code can also be accessed from the same source under https://zenodo.org/records/11058350 and can be run using python.
The software is published under the GNU public license.

*Author contributions.*   M.B provided the base code for the calibration/digitization modeling and E.M gave us access to the paper spectra. J.N,
M.P.C, M.P and E.M supervised this work. J.M further developed the Digitization/Calibration software, scanned, digitized, and calibrated
the spectra.

*Competing interests.*   At least one of the (co-)authors is a member of the editorial board of Atmospheric Measurement Techniques.



*Acknowledgements.* We are grateful to the staff of the University of Liege who have kept these historical spectra in relatively good condition
throughout the years. The simulated spectrum was produced using SFIT4, which is one of the first advanced atmospheric retrieval algorithms
developed through the collaboration between the University of Colorado and the University of Bremen. The software utilizes a line-list
generated by HITRAN. This work was funded by the Deutsche Forschungsgemeinschaft (DFG, German Research Foundation) within the
projects NO 404/24-1 and PA 1714/8-1





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
