# Peer review of "Digitization and calibration of historical solar absorption infrared spectra from the Jungfraujoch site"

_Atmospheric Measurement Techniques, 2024_

## Author Response (AR1)

**We thank both reviewers for their valued comments and we offer our answers to the issues they raised:**

**Reviewer 1:**

L4: "microns (~421 to 3571cm−1 ) that later contributed to the development of the HITRAN database." rather: "microns (~421 to 3571cm−1 ) that *in particular* later contributed to the development of the HITRAN database"

L4 **Answer**:

Done

L4: "We now digitized these old… rather: "We now digitized these analog recorded…

L4: **Answer**

Done

L17: technologies (Zander et al., 2008). Its strategic location at high altitude, coupled with minimal interference from pollution and… Are there other instruments a the JFJ / Sphinx that are no described in Zander 2008? Should there be other references?

L17 **Answer**:

Following the grating spectrometer in the 1950s, another grating instrument was installed later in the 1950s, regularly updated and it provided regular measurements until the late 1980s. In addition, a homemade FTIR was developed and tested before starting regular operations in 1984. From the early 1990s, an FTIR manufactured by the German company Bruker (IFS 120HR) and modified by the U.Liege team was added and operated in parallel with the homemade spectrometer. The homemade FTIR was later retired in 2008 (Zander et al., 2008), while the Bruker instrument has been in use until Summer 2024 (Prignon et al., 2019). It has now been replaced by a IFS125HR instrument, also from Bruker. We now mention this in the manuscript (line 30)

L21: "from the late 1950 to 1980 a seven" rather: "from the late 1950's to 1980 a seven…

L21 **Answer**:

Done

L22: "near UV near IR" rather: "near UV to near IR…

L22 **Answer**:

Done

L30: "covering a broad range of wavenumbers… rather: "covering a wide spectral range…

L30 **Answer**:

Done

L35: "through visual comparison with high resolution spectra. > which high resolution spectra?

L30 **Answer**:

The comparison was performed visually by examining spectral lines from 1951, as well as the ones recorded by the homemade FTIR spectrometer and the spectra measured by a Bruker 120 HR FTS at 0.0035 cm 1 resolution under similar solar zenith angle conditions in the year 2000 (Zander et al., 2008). We now mention this in the manuscript (line 40).

Fig 1b. Was the instrument using a multipass cell in particular? There are multi pass cell Pfund types but not all are multi cell. An accurate diagram here is important.

Fig1b **Answer**:

Unfortunately information on the original design of the instrument is lost. But we believe there was no multipass cell in the beam. A multipass cell installed in the lightpath would need to be removed before solar absorption spectra could be taken. The schematic Figure has been modified to remove the multipass cell label.

L104: "This gives us a relative mean digitization error of about: $\varepsilon = (9.3 * 100)/600 = 1.55\%$". If there are 9.3 pixel uncertainty in a page not an inch. So a page (or ordinate scale) of (for in stance) 5 inches has 3000 pixels so the nominal error is ~x5 less?

L104 **Answer**:

The error that we calculated was related to the image resolution of the scanner itself which is 600dpi, independent of the dimension of the image given. This was proved in the validation sec tion where the residual, when compared to an original spectrum, also gave a difference of 1.55%. However, the error per image at an ordinate scale is in fact approximately 5 times lower than the error when taking the full height of a given image. We modified the text to make this clearer. (line 110)

L121: "The choice of the apodization function for the interferograms is connected to the instru mental properties of a grating spectrometer, since for a diffraction limited resolution, where the image of the spectral line is no longer the same shape as the entrance slit but controlled by the diffraction pattern, the instrumental line function in the spectral domain corresponds to a sinc2 func "

This description should be more detailed or better yet left out and simply use a reference. Devel opment or explanation of a grating instrument function is not important here.

L121 **Answer**:

We agree that a detailed discussion is not necessary here. We now simply write that the ILS is determined by the limited resolution of a grating, leading to a sinc2 ILS function, plus the effect of the aperture size, leading to a triangle ILS function. In our case, due to the detectors at that time with their low SNR, the aperture dominates, leading to a triangle ILS function. (line 130)

L 126: Since the spectral resolution variation of the grating wrt wavenumber is known why was this variation not used? Or discrete smaller sections of the spectrum fitted with slight increasing (or decreasing) resolutions.

L126 **Answer**:

The wavenumbers given on the original plots were not very precise. Therefore we decided to use specific wavenumber points from synthetic spectra to get the wavenumber scale. This has been checked for larger spectral regions, the fitted spectra match the digitized ones very well.

L133: its not clear what specific calibration is being performed. There is baseline, spectral resolution, wavenumber scale, 100% transmission to be considered.

L133 **Answer**:

We used the Least Squares fitting lmfit library, which performs a fit of chosen pixel points to known wavenumber values. The algorithm adjusts the digitized spectrum to match the new wavenumber scale instead of the wavelength scale using the fitting equation $y = ax^2 + bx + c$ where x is the pixel scale the y is the new wavenumber scale. During the calibration the spectra's baseline was left unchanged (since it can be fitted in many cases during the trace gas retrieval). The wavenumber scale was fitted using known calibration points from a known synthetic spectrum, as explained above (Line 140 and Line 155).

what is recorded are transmission spectra, so we don't know the 100% background, and for the retrieval we do not need it. There might be effects from filters, grating, detectors that influence the background, but we retrieve small spectral windows, where we can assume that, while the background level is unknown, it is constant. In order to account for small background variations, we sometimes fit the slope (Line 160).

L147: Is this the SFIT fitting or some other fitting technique using the sfit generated spectrum? What parameters are being fitted? Perhaps a defined equation would be clearer.

L147 **Answer**:

See L113. As explained in L133 the calibration is not a trace gas retrieval algorithm, it is performed using a simple 2nd degree polynomial fitting of chosen pixel values and matching wavenumber values in a synthetic spectrum to find the appropriate wavenumber scale of the previously digitized spectrum. This is performed using the python library lmfit . We now rephrased it to make it more clear. (Line 155)

We rephrased the text and together with the changes L113 (above) it should be clear now.

L150: "saved as a text format… rather "saved in a text format file…

L150 **Answer**:

Done

L173: Its not clear what this description is for? Do we not believe PVLIB?

L173 **Answer**:

We compared the calculation of the solar zenith angle using a different calculation algorithm to be more rigorous and verify the correctness of our calculated solar zenith angle. We agree that it might not have been necessary, since PVLIB is a widely used and accurate algorithm. We shortened this discussion in the manuscript.

For a record time of ~1.5h and more : what time was used for SZA / airmass calculation?

**Answer**:

The solar zenith angles were calculated each minute and saved as an array in the text file. The start and end time were noted by the original authors on the historic spectra. The decision which SZA to be used for an airmass calculation in a trace gas retrieval is left to the users, we provide the range of SZA relevant for a certain spectral window.

The term detection boundary and detection limit are used interchangeably perhaps better is the term 'peak threshold' would be clearer, less ambiguity, that we are talking about the ordinate region where peaks are determined.

L173 **Answer**:

The algorithm to detect the peaks for calibration need a parameter called prominence, which defines how much a peak stands out from the surrounding baselines. This term is added by the authors to the text instead of peak threshold to avoid any ambiguity.

**Reviewer 2:**

It would be great to have a larger image of a scanned spectrum, including pen artifacts, hand written notes, adhesive tape etc.. The one shown on Fig. 2a is very small.

**Answer:**

Done. We now show a larger region.

If I understood correctly, the term "line thickness" is sometimes used in confusing ways. I would suggest to use "line thickness" for the thickness of the line drawn by the pen and "line width" for the spectral line width. This is especially relevant for Sec. 3.1. Please provide a definition for these two terms at the beginning of that section.

**Answer:**

We fully agree with this comment. The text has been modified accordingly to avoid confusion.

I missed a section on data availability at the end of the manuscript. I guess this will be added for the final version?

A link to download the data has been added to data availability. As a general comment, we added the following references during the revision phase:

Prignon M, Chabrillat S, Minganti D, O'Doherty S, Servais C, Stiller G, et al. Improved FTIR retrieval strategy for HCFC 22 (CHClF2), comparisons with in situ and satellite datasets with the support of models, and determination of its long term trend above Jung fraujoch. Atmospheric Chemistry and Physics 2019;19:12309–24. ht tps://doi.org/10.5194/acp 19 12309 2019.

Zander R, Mahieu E, Demoulin P, Duchatelet P, Roland G, Servais C, et al. Our changing atmosphere: Evidence based on long term infrared solar observations at the Jungfraujoch since 1950. Science of The Total Environment 2008;391:184–95. ht tps://doi.org/10.1016/J.SCITOTENV.2007.10.018.